# VPg Impact on Ryegrass Mottle Virus Serine-like 3C Protease Proteolysis and Structure

**DOI:** 10.3390/ijms24065347

**Published:** 2023-03-10

**Authors:** Gints Kalnins, Rebeka Ludviga, Ieva Kalnciema, Gunta Resevica, Vilija Zeltina, Janis Bogans, Kaspars Tars, Andris Zeltins, Ina Balke

**Affiliations:** 1Structural Biology Group, Latvian Biomedical Research and Study Centre, Ratsupites Street 1, k-1, LV-1067 Riga, Latvia; 2Plant Virus Protein Research Group, Latvian Biomedical Research and Study Centre, Ratsupites Street 1, k-1, LV-1067 Riga, Latvia; 3Plant Virology Group, Latvian Biomedical Research and Study Centre, Ratsupites Street 1, k-1, LV-1067 Riga, Latvia; 4Biotechnology Core Facility, Latvian Biomedical Research and Study Centre, Ratsupites Street 1, k-1, LV-1067 Riga, Latvia

**Keywords:** ryegrass mottle virus, serine-like 3C proteases, VPg, *Sobemovirus*

## Abstract

Sobemoviruses encode serine-like 3C proteases (Pro) that participate in the processing and maturation of other virus-encoded proteins. Its *cis* and *trans* activity is mediated by the naturally unfolded virus-genome-linked protein (VPg). Nuclear magnetic resonance studies show a Pro–VPg complex interaction and VPg tertiary structure; however, information regarding structural changes of the Pro–VPg complex during interaction is lacking. Here, we solved a full Pro–VPg 3D structure of ryegrass mottle virus (RGMoV) that demonstrates the structural changes in three different conformations due to VPg interaction with Pro. We identified a unique site of VPg interaction with Pro that was not observed in other sobemoviruses, and observed different conformations of the Pro β2 barrel. This is the first report of a full plant Pro crystal structure with its VPg cofactor. We also confirmed the existence of an unusual previously unmapped cleavage site for sobemovirus Pro in the transmembrane domain: E/A. We demonstrated that RGMoV Pro *in cis* activity is not regulated by VPg and that *in trans*, VPg can also mediate Pro in free form. Additionally, we observed Ca^2+^ and Zn^2+^ inhibitory effects on the Pro cleavage activity.

## 1. Introduction

Ryegrass mottle virus (RGMoV) belongs to the family *Solemoviridae*, genus *Sobemovirus*, and infects monocotyledons, such as Italian ryegrass (*Lolium multiflorum*), cocksfoot (*Dactylis glomerata*), wheat (*Triticum aestivum*), barley (*Hordeum vulgare*), oats (*Avena sativa*), rye (*Secale cerdeale*), and Italian millet (*Setaria italica*) [1,2]. RGMoV was isolated from Italian ryegrass and cocksfoot in Japan [1] and was also found in irrigation waters in Ecuador in 2021 [3]. Its genome is a monocistronic single-stranded positive-sense RNA that is 4212 or 4247 nucleotides in length [3,4]. Sobemovirus genomic RNA (gRNA) encodes five open reading frames (ORF), where ORF1, ORFx, ORF2a, and ORF2b are translated from the gRNA, and ORF3, which encodes the coat protein (CP), is translated from a subgenomic RNA (sgRNA) [5,6]. gRNA and sgRNAs lack a cap structure at the 5′-end and a poly(A)-tail at the 3′-end [7]. Instead, the 5′-end is protected by covalently bound VPg (mimicking eukaryotic cap structure) in RGMoV, connecting the OH group of the first serine (S) and a terminal 5′-phosphate group of the RNA genome via a phosphodiester bond [8]. ORF2a and ORF2b of sobemoviruses encode two polyproteins, where the *N*-terminal parts of ORF2a and ORF2ab encode serine-like 3C protease (Pro), which displays a significant similarity to the cysteine (C) proteases of picornaviruses [9]. The next domain encoded by both ORFs is VPg, followed by P18, which can be divided into two domains: RNA-binding (P10) and an ATPase domain (P8) in the *C*-terminal part of ORF2a [10]. The *C*-terminal part of polyprotein P2ab contains an RNA-dependent RNA polymerase motif (RdRp). In sobemoviruses, the location of VPg between Pro and RdRp is unique among phylogenetically related polero-, enamo-, and barnaviruses [11]. Functional domains from polyproteins of sobemoviruses are released by proteolytic processing by Pro [8,12,13]. Pro catalyzes the hydrolysis of peptide bonds located between two specific residues, the first of which is glutamic acid (E). Identification of sobemoviral VPgs attached to the viral genomes indicated that the 2a polyprotein was processed for southern bean mosaic virus (SBMV) at E/T (glutamic acid/threonine) residues at positions 325/326 and 402/403, cocksfoot mottle virus (CfMV) at E^317^/N^318^ (glutamic acid/asparagine) and E^395^/T^396^ residues, rice yellow mottle virus (RYMV) at E^326^/S^327^ and E^404^/T^405^ residues, and RGMoV at E/S residues at positions 314/315 and 393/394 [8,13]. VPg can directly regulate Pro activity, as Sesbania mosaic virus (SeMV) Pro is active *in trans* only in fusion with VPg [14]. Mutagenesis analysis revealed that residue W^43^ (tryptophan) of VPg interacts with SeMV Pro residues W^271^ and H^275^ (histidine), activating Pro [15]. VPgs of plant viruses are characterized as “natively unfolded proteins”, lacking a well-defined 3D structure and existing in multiple conformations at physiological conditions [16]. These proteins adopt a rigid conformation stabilized in vivo upon interaction with natural substrates and are multifunctional [11]. Bioinformatics methods have unfolded the nature of VPgs of several sobemoviruses (SeMV, RGMoV, CfMV, RYMV, SBMV, and southern cowpea mosaic virus (SCPMV)), potyviruses (lettuce mosaic virus (LMV), potato virus Y (PVY), potato virus A (PVA), tobacco etch virus, turnip mosaic virus, bean yellow mosaic virus), and *Caliciviridae* family members (rabbit hemorrhagic disease virus (genus *Lagovirus*), vesicular exanthema of swine virus (genus *Vesivirus*), sapporo virus Manchester virus (genus *Sapovirus*), and Norwalk virus (genus *Norovirus*)) [14,16,17,18]. In VPgs of the RYMV group (to which RGMoV belongs), similar to potyvirus and *Caliciviridae* family members, the *N*- and *C*-termini are predicted to be disordered [16,18]. Secondary structure prediction in this sobemovirus group indicated the presence of an α-helix, followed by two β-strands, and another α-helix. Some of the terminal regions of these VPgs are predicted to have propensities that are both disordered and folded into α-helices. Residues 48 and 52, which are associated with RYMV virulence, are located in the *C*-terminal region [19]. Recent nuclear magnetic resonance (NMR) studies of SeMV VPg have demonstrated that it has a novel tertiary structure with an α-β-β-β topology [20]. Circular dichroism (CD) spectrum analysis for SeMV, RYMV, LMV, PVY, and PVA also revealed features typical of intrinsically disordered proteins [14,16,18,21]. NMR analysis of VPgs was also performed for PVY, PVA, and cowpea mosaic virus [18,21,22]. PVA VPg behaves as a partially folded species that contains a hydrophobic core domain [21]. Currently, the 3D structures of VPg peptides are known only for viruses infecting mammals. Determination of the 3D structure of the first 15 amino acids (aa) of the *N*-terminus of foot-and-mouth disease virus (family *Picornaviridae*) VPg1 was possible only with RdRp, and the *C*-terminus remained completely disordered [23,24]. A similar structure was observed for enterovirus 71 (family *Picornaviridae*) VPg, where the first 20 aa formed a V-shaped extended conformation from the front side of the catalytic center to the back of RdRp [25]. The 3D structure of coxsackievirus B3 (family *Picornaviridae*) VPg 7–15 aa [26], the 3D structure of the feline calicivirus (FCV), and murine norovirus (MNV) VPg core of the protein adopts a compact helical structure flanked by flexible *N*- and *C*-termini. The core (10–76 aa) of FCV VPg (10–76 aa; PDB ID 2M4H) contains a well-defined three-helix bundle, and the MNV VPg core (11–85; PDB ID 2M4G) contains only the first two of these secondary structure elements [27]. The same structure remains during the interaction with the palm domain of the MNV RdRp [28].

Sobemovirus Pro is a glutamyl endopeptidase (GEPases). GEPases are serine proteases that preferentially cleave peptide bonds following E [29,30]. GEPases have been identified and characterized in gram-positive bacteria and (+)RNA viruses. All GEPases belong to the chymotrypsin structural family, one of the most extensively studied enzyme families. Chymotrypsin-like protease (CLP) molecules share their spatial organization principle, the so-called chymotrypsin (or trypsin) fold [31], which comprises two β-barrels with an active site located between them. The only 3D structure of the sobemovirus Pro domain available in the worldwide Protein Data Bank (PDB) is that of SeMV Pro, determined at a resolution of 2.4 Å (PDB ID 1zyo). Its 3D structure has a typical CLP fold with a well-formed active site and a substrate-binding cleft that is more similar to that of cellular [in particular, Glu-specific protease from *Streptomyces griseus* (Glu–SGP; PDB ID 1 hpg)] rather than viral Pro [12,32]. Pro carries the conventional D-H-S catalytic triad, and modification of its residues terminates polyprotein processing [12]. The consensus aa sequence around the catalytic triad is H(X^32–35^)[D/E](X^61–62^)TXXGXSG [33]. Similarly, for all GEPases, the conserved residues of SeMV Pro H^213(298)^ and T^190(279)^ are maintained, located within the S1 site of the enzyme. However, position 216(301) is occupied by a large hydrophobic residue, phenylalanine (F; [31]). SeMV Pro comprises two β-barrels connected by a long interdomain loop. Both the active site and substrate-binding cleft are located between the two barrels and are fairly exposed to a solvent. Mutation analysis of the glutamate-binding site (S1-binding pocket) residues H^298^, T^279^, and N^308^ of SeMV Pro demonstrated that these are indeed crucial for Pro activity. In addition, several downstream residues are shown to be important for Pro activity [32]. The substrate specificities of SeMV Pro were predicted to be N, Q-E/T, and S-X (where X is an aliphatic residue) [10]. However, multiple sequence alignments revealed no common substrate specificity for all sobemoviruses, except that the S1 binding pocket seems to be highly specific for glutamate (Glu) or glutamine (Q) [34].

In this study, we demonstrated that a 10-residue-long VPg peptide in the *C*-terminal region (GEMTWADMVE) can assume a helical conformation and bind to a specific site in the Pro domain, causing a conformational change. We also confirmed an additional Pro cleavage site that has not been mapped for sobemoviruses after a transmembrane domain (TMD) between E/A (glutamic acid/alanine) residues, and demonstrated that *in trans* Pro activity can be induced by non-covalent protein–protein interactions with VPg, without the requirement of fusion to the Pro domain, as was reported previously [14].

## 2. Results

### 2.1. Description of Serine-like 3C Protease, Its Fusion Variant with Cofactor VPg, and the Catalytic Triad S Mutant Used for Structure Determination

An RGMoV cDNA clone (GenBank No. EF091714, [4]) was used as a DNA template to amplify the *N*-terminal truncated Pro coding domain (Δ50Pro) lacking 50 residues of TMD (counting from the first methionine (M), predicted by the TMHMM program [35]), Δ117Pro catalytic triad S^159^ mutant (Δ117Pro^cm^, S159A), and Δ50Pro fusion version with VPg (Δ50Pro-E/A-VPg), where both predicted Pro cleavage sites between Pro and VPg domains were mutated—E^315^ and E^318^ to A (E315A and E318A)—to prevent VPg processing from Pro. The constructs contained an *N*-terminal 6xH-tag for purification by Ni^2+^-immobilized metal affinity chromatography (IMAC). After expression and protein purification on IMAC and sample analysis by sodium dodecyl-sulfate polyacrylamide gel electrophoresis (SDS-PAGE), we observed that the molecular weights (MWs) of the Δ50Pro and Δ50Pro-E/A-VPg products were lower than expected. Instead of 29.5 kDa for Δ50Pro, it was 20.9 kDa, and for Δ50Pro-E/A-VPg, while the predicted MW was 38.5 kDa, the observed MW was 29.5 kDa (Figure 1B).

Proteolysis occurred at the *N*-terminal residues because the proteins were unable to bind to the IMAC column (Figure 1B). A similar situation was previously observed in the case of SeMV Δ70Pro [32] and in our previous experiments [17]. However, the expressed RGMoV Pro sequence did not contain any confirmed Pro cleavage sites typically used by sobemoviruses [11]. Δ50Pro, Δ50Pro-E/A-VPg, and Δ117Pro^cm^ were purified by gel filtration on a Superdex200 column for further structural studies and measurement of the MW of the cleavage product for localization of the cleavage site. To identify the exact proteolytic site, we performed mass spectrometry (MS) analysis of the purified Δ50Pro *C*-terminal domain. MS analysis identified a 20.93 kDa protein (Figure 1C) that corresponded to the protein band visible in SDS-PAGE (Figure 1B). Based on the MS data, we concluded that the protein is 67 residues shorter, and that proteolysis could occur between residues 117 and 118, resulting in a Pro domain with a total truncation of 117 aa (Δ117Pro). The predicted MW of Δ117Pro^cm^ was 21.6 kDa, but by MS analysis, it was 21.4 kDa (Figure 1C). This demonstrates that Δ117Pro^cm^ lacks the first M because of the following small residue [36]. A similar pattern was observed for bacteriophage Qβ CP expressed in yeasts [37] and two distinct sobemovirus CPs (CfMV and RYMV) expressed in *Escherichia coli* [38].

### 2.2. Identification of Serine-like 3C Protease Transmembrane Domain Cleavage Site

Residues 117 and 118 in Pro correspond to E and A, respectively; therefore, we concluded that cleavage occurred at the E^117^/A^118^ site. To confirm the presence of an E/A site in RGMoV, we mutated E^117^ to A (E117A) for both Δ50Pro and Δ50Pro-E/A-VPg, leaving a 6xH-tag at the *N*-terminus (Figure 2, No 4 and 5).

After expression and subsequent purification by IMAC, we purified the Δ50Pro (29.5 kDa) or Δ50Pro-E/A-VPg (38.5 kDa) domains of the expected lengths, which were present in the IMAC elution fractions (Figure 3A,B). This confirmed that proteolytic cleavage was indeed prevented by mutating the E^117^/A^118^ site, and no additional cleavage site was used, even though the Pro aa sequence contains several corresponding cleavage sites [4]. In addition, we observed that the expression levels of Δ50Pro-TMD-N6H and Δ50Pro-TMD-E/A-VPg-N6H were reduced compared to those of Δ50Pro-N6H and Δ50Pro-E/A-VPg-N6H. Δ50Pro-TMD-E/A-VPg-N6H was less soluble (Figure 3A,B, line T, S, P) which could be explained by the importance of the E^117^/A^118^ site in Pro maturation.

The expression of Δ23Pro with the *C*-terminal 6xH-tag was low (32.4 kDa; Figure 2, No 15; Figure 4), and the protein was mostly insoluble, but proteolysis *in cis* was observed by purification of Δ117Pro-C6H in small amounts (21.75 kDa; Figure 2, No 15; Figure 4A, line E1), which was detected by Western blotting (WB) with antibodies against a 6xH-tag (Figure 4B, line E1).

The alignment of RGMoV Pro *N*-terminal part with 20 sobemoviruses Pro sequences demonstrated that the E/A cleavage site is also present in other sobemoviruses, including Rottboellia yellow mottle virus, RYMV, Imperata yellow mottle virus, Cymbidium chlorotic mosaic virus, Cymbidium chlorotic mosaic virus, and lucerne transient streak virus, which confirms the possibility of utilizing other cleavage site variants [34]. Alignment also revealed other possible cleavage site sequences. In Artemisia virus A and velvet tobacco mottle virus, we identified an E/G (glutamic acid/glycine) cleavage site. Additionally, we identified an E/C site in four cases, papaya lethal yellowing virus, Rubus chlorotic mottle virus, sowbane mosaic virus, and subterranean clover mottle virus, and an E/M site in CfMV (E/T and E/N cleavage sites is in close proximity [34]), but none of them were tested in vivo. In the remaining sobemoviruses, we identified previously described cleavage sites, such as E/S, in five viruses, namely SBMV, SeMV, Solanum nodiflorum mottle virus, soybean yellow common mosaic virus, and SCPMV, and E/T in turnip rosette virus and blueberry shoestring virus (Appendix A). Considering the controversial results from SeMV Pro that we obtained, we were eager not only to test Pro activity *in cis* and *trans* in different combinations with ORF2a encoded proteins, but also to confirm the proteolytic cleavage sites used for RGMoV ORF2a.

### 2.3. Confirming of ORF2a Proteolysis Cleavage Sites

To confirm ORF2a and to test the *E. coli* system for Pro cleavage site identification and confirmation, we created two expression vectors. One contained a native Δ50Pro-VPg fusion with 6xH-tag at the *C*-terminus of VPg (Figure 2, No 6) and Δ50Pro-E/A-VPg fused with the P16 domain (equivalent to the SeMV P18 domain) with 6xH-tag at the P16 *C*-terminus (Figure 2, No 7). SDS-PAGE analysis revealed the cleavage of separate VPg-C6H or P16-C6H domains, which were subsequently purified by IMAC (Figure 5A,B).

Elution fractions were additionally purified by gel filtration on a Superdex200 column for analysis with MS to confirm the MW of proteins, because VPg and P16 showed abnormal behavior in SDS-PAGE, and to identify Pro cleavage sites (Figure 5A,B). A similar situation was observed for CfMV VPg, as it was isolated from the host organism and detected in WB as a 24 kDa protein and not a 12 kDa protein, as expected [39]. The shift in mobility in SDS-PAGE could be caused by the acidic nature of VPg proteins (IP~4). This abnormal mobility has been reported previously [40]. A similar behavior of the P16 homolog has been reported earlier [17]. This unusual behavior of P16 in SDS-PAGE could be explained by its high isoelectric point (IP = 9.26). Positively charged proteins can also behave anomalously on SDS-PAGE, as in the case of the SeMV P8 protein [41]. VPg-C6H MS analysis of MW confirmed that in VPg, the E/S cleavage site was used. This has also been described for another native VPg [13]. The predicted MW of cleaved VPg-C6H MW was 9.87 kDa as revealed by MS analysis (MW 9.85 kDa; Figure 5C), which is in good agreement with the predicted MW and E/S sites. In the P16-C6H case, MS analysis confirmed the usage of the E/S site (E^393^/S^394^). To confirm this and to exclude any possibility of an additional preferable cleavage site, as was described for SeMV [41], we mutated the cleavage site E^393^ to A between the VPg-P16 domains (E393A; Figure 2, No 10). Co-expression with Δ50Pro-N6H analysis by SDS-PAGE did not show any cleavage of VPg from P16 (Figure 6C), thereby confirming the previously identified cleavage site. In addition, no additional processing products that could be confirmed were detected. Therefore, this raises the question of whether the P18 domain of SeMV undergoes processing *in cis* [41]. As demonstrated in our previous experiments, RGMoV P16 tends to oligomerize and can be classified as a natively unfolded protein [17]. This experiment also revealed that the *E. coli* expression system is acceptable for Pro cleavage site mapping and testing as a fast and inexpensive preliminary method.

### 2.4. Serine-like 3C Protease trans- and cis-Activity Tests in the Presence and Absence of VPg

We were curious if RGMoV Pro was active only *in trans* when it was in fused with VPg, as described for SeMV. To test our hypothesis, we tested the *in cis* and *trans* activities of Pro in a two-plasmid system. For the *in trans*-cleavage test, we expressed Δ50Pro as a single domain from the pCold1 plasmid, and a VPg-P16 domain with a 6xH-tag at the *C*-terminus of P16 from the pET28a(+) plasmid (Figure 2, No 9). Total cell lysate analysis revealed that Pro could cleave the VP-P16-C6H fusion domain (Figure 6A) into VPg and P16-C6H, showing a similar pattern in SDS-PAGE as in the case of Δ50Pro-E/A-VPg-P16-C6H, where the protein with a MW corresponding to that of P16-C6H was purified by IMAC (Figure 5B). As a control, we expressed the VPg-P16-C6H fusion protein separately (Figure 2, No 8; Figure 6B) and its E393A mutant in co-expression with Δ50Pro (Figure 2, No 10; Figure 6C). These results indicate that RGMoV Pro does not lose its *trans* and *cis* activity when VPg is expressed as a separate domain. This is in contrast to previous in vitro experiments, which showed that Pro is active only when VPg is fused to its *C*-terminal part [14]. This indicated that the RGMoV Pro cleavage mechanisms were different.

In the SeMV Pro E133A mutant, Nair and colleges proposed that its activity is reduced, and P18 processing into P10 and P8 is corrupted [41]. Therefore, we performed a Pro activity test by co-expressing Δ50Pro-TMD-N6H with VPg-P16-C6H (Figure 2, No 12; Figure 6E), and created a Δ50Pro-TMD fusion construct containing a mutated cleavage site between the Pro and VPg domains and active sites between the VPg and P16 domains and the 6xH-tag at the *N*- and *C*-termini (Δ50Pro-TMD-E/A-VPg-P16-NC6H; Figure 2, No 11; Figure 6D). As revealed by SDS-PAGE analysis, Pro was still active *in trans*, cleaving off separate P16-C6H domains in both cases, but in a reduced manner (Figure 6D,E), highlighting the importance of the appropriate matured form of Pro during polyprotein processing in terms of speed.

In addition, in this case, no P16 processed variant was observed. In the SeMV case, it was possible to identify a weak signal of the 14 kDa protein in WB as the *N*-terminal part of ΔN70Pro with 6xH-tag, which could be recognized by polyclonal antibodies raised against P8 with 6xH-tag [41]. As revealed by SDS-PAGE analysis, in RGMoV, Δ50Pro with the 6xH-tag *N*-terminal part after cleavage in SDS-PAGE migrates abnormally as a ~15 kDa protein instead of an 8.6 kDa protein (Figure 1B). This was due to its acidic nature (IP–4.61). The SeMV Pro *N*-terminal part is also acidic (IP–4.29), and its MW (7 kDa) is close to the mass of P8; therefore, it is difficult to confirm that it is indeed P8. These in vivo experiments revealed that the RGMoV *C*-terminal domain after VPg does not undergo processing into two separate domains, despite several Pro cleavage sites being used *in trans* or *cis* [4], and our previous experiments confirmed P16 as a monolithic domain [17].

To test Δ50Pro *cis* activity in the absence of VPg, we expressed Δ50Pro with a cleavage site before the 6xH-tag sequence at the *C*-end (Figure 2, No 13). SDS-PAGE analysis revealed that Pro is active *in cis*, releasing Δ117Pro, which can be purified by IMAC (Figure 7A). In addition, Δ117Pro released from Δ50Pro, used in 3D determination and MS analysis, confirmed the presence of an *in trans* cleavage site with two additional aa—ESSN—in the Pro *C*-terminal region (Figure 1). This confirmed that Pro has *in cis* activity without VPg.

This construct was also used for co-expression with VPg to test for *trans* activity. Co-expression analysis of total cell lysates revealed an additional protein band (Figure 7B). After sample analysis of the purified proteins by IMAC, the newly appeared protein band did not bind to the column material. This confirms that the *in trans* activity of Pro is mediated by the presence of VPg and the importance of protein–protein interactions.

The expression of the Pro^cm^ variant confirmed the importance of S^159^ for the catalytic activity of RGMoV Pro (Appendix A).

### 2.5. Crystal Structures of Δ117Pro, Δ117 Pro-E/A-VPg, and Δ117Pro^cm^

We crystallized and solved 3D crystal structures for two catalytically active forms of Δ117Pro: one in its VPg-free form, the second in its VPg-bound state with fused VPg at the *C*-terminal part (Δ117Pro-E/A-VPg), and one catalytically defective Δ117Pro^cm^ form with S^159^/A mutation. The overall fold change of Δ117Pro is typical for CLP and is very similar to the structure of SeMV Pro (Figure 8).

The VPg sequence GEMTWADMVE, corresponding to residues 256–265 in Δ117Pro-E/A-VPg, forms a short α-helix that binds to the surface of Δ117Pro (Figure 9A). W^260^ is a key residue in this interaction; it is located in a very tight and largely hydrophobic pocket, forming a hydrogen bond with the main-chain carbonyl oxygen of D^77^. The interaction of the VPg helix with Δ117Pro is also ensured by two additional hydrogen bonds: one between the K^52^ side chain nitrogen and V^264^ main chain oxygen, and the second between the I^82^ main chain nitrogen and M^258^ main chain oxygen (Figure 9B). VPg-binding sites are mostly hydrophobic, as they are formed by L^51^, M^55^, M^76^, W^80^, I^82^, L^87^, I^94^, and V^96^ (Figure 9C). The VPg helix is somewhat amphipathic, having W^260^ and V^264^ side chains located towards the VPg site, and T^259^, D^262^, and E^265^ side chains exposed to the outside environment, so the VPg–Pro interaction is also strengthened by hydrophobic interactions. Curiously, SeMV Pro contains an analogous hydrophobic pocket, but it very likely does not have a similar function, mainly because W^183^ already occupies its center (Figure 9D).

The binding of VPg causes several significant conformational shifts in the Δ117Pro structure (Figure 4C and Figure 5).

First, in the absence of VPg, L^87^ shifted towards the VPg-binding pocket. Consequently, the eIb β-strand is also pulled towards the VPg binding site, disrupting its β-strand conformation and significantly altering the position of the loop between the eIb and fI strands. Second, the α1 helix in the VPg-free structure is also slightly shifted towards the empty VPg binding site (Figure 9C); although, the position of the catalytic triad H^49^, located on this helix, is minimally affected. These two conformational shifts are driven by the presence or absence of VPg, since they align well in the Δ117Pro^cm^ and Δ117Pro cases, which do not contain VPg at all.

There were several other conformational shifts for which no clear causation between the presence and absence of VPg could be ascertained. The rotation of the catalytic triad D^92^ (Figure 9C) is one such feature. In Δ117Pro chain A, it is rotated away from the catalytic triad H^49^, thereby disrupting the functionality of the catalytic triad since D^92^ is too far from H^49^ to ensure any proton transfer. However, in the Δ117Pro chain B, Δ117Pro^cm^, and Δ117Pro-E/A-VPg cases, D^92^ is directed towards H^49^, as would be expected in an active Pro. There are several more conformational shifts unique only to Δ117Pro (Figure 10) and that are not present in Δ117Pro^cm^ and Δ117Pro-E/A-VPg. These are the partial unwinding of the βII barrel and subsequent movement of the bII and cII β-strands and the cII/dII loop, and the disordering of the eII/fII loop (Figure 10A). These conformational shifts may affect the positioning of G^157^ (Figure 10B). This residue, judging by comparison with the 3D structures of other serine Pros, is responsible for the formation of the oxyanion hole, and its disruption in the Δ117Pro structure could directly modulate the catalytic activity. These conformational shifts also had a significant influence on the position of conserved key residues forming the Glu-binding site (H^150^, H^173^, T^154^, and N^183^; Figure 10C,D). While H^173^ and N^183^ did not change their positions significantly, H^150^ and T^154^ were noticeably shifted due to the movement of bII-cII strands. This shift could also render the Glu-binding site unsuitable for cleavage site recognition.

None of these conformational shifts were observable in SeMV Pro; both the catalytic triad and Glu-binding regions aligned almost perfectly with Δ117Pro-E/A-VPg (Figure 11). Such differences can be partially explained by the C^277^-C^248^ disulfide bond in the SeMV Pro structure (Figure 11). This disulfide anchors the cII strand and does not allow for movement as in the Δ117Pro case. Curiously, RGMoV contains one C in this position, C^152^, but the other corresponding residue is A^124^, so there is no comparable stabilizing influence.

Nevertheless, the C^277^-C^248^ disulfide bridge was demonstrated for SeMV Pro as non-essential for catalytic activity; therefore, its absence could be relatively unimportant [12,32]. For RGMoV Pro, there is no intermolecular disulfide bond present in SeMV Pro, and, as a result, the former is a monomer. This former SeMV intramolecular disulfide could also be unimportant because it has been speculated to be an artifact of crystallization [32]. However, RGMoV Δ117Pro has an internal C^38^-C^195^ disulfide bridge that is not present in SeMV Pro. This illustrates that the Pro domain disulfide bridges, although potentially important for conformations, are not particularly conserved among different viruses. Potential disruption of the RGMoV Pro internal C^38^-C^195^ disulfide bridge by 20 mM dithiothreitol (DTT) led to the inactivation of proteolytic activity (Appendix A), so the importance of this particular bridge in the activity of RGMoV Pro remains open.

## 3. Discussion

Sobemoviral Pros are known to cleave E/T, E/S, and E/N residues [39,41,42], but all cleavage sites are characterized only for SeMV [41]. Only cleavage sites between Pro and VPg were identified for CfMV, SBMV, RYMV, and RGMoV [8,13,39]. Sequence comparison of TMD Pro cleavage sites revealed the presence of an E/A cleavage site, which was confirmed by site-directed mutagenesis in RGMoV. Six more sobemoviruses were identified that could use the same cleavage site and probably also the E/C, E/G, and E/M sites (Appendix A). The VPg-P16 domain in RGMoV was processed using an E/S cleavage site identical to that of the Pro-VPg site. However, we did not observe further processing of the P16 domain into two subdomains, as described previously for SeMV [41], and we propose that it is not processed in *E. coli* in the SeMV case. The identified protein in WB could be the Δ70NPro *N*-terminal domain containing 6xH-tag as P8, against which polyclonal antibodies were raised. Our experiments revealed that a bacterial expression system combined with MS analysis and site-directed mutagenesis is suitable for the identification of Pro cleavage sites. In addition, this system allows the testing of both *in cis* and *trans* Pro activities. Pro activity *in trans* can be achieved with free VPg co-expression without fusion, as described previously [14]. Unfortunately, we were unable to develop a FRET system for Pro catalytic activity *in cis* to test and evaluate VPg’s impact on the catalytic speed. However, we were able to detect peptide cleavage by robust MS analysis (Appendix A). Additionally, we performed an in vitro cleavage test using ∆50Pro^cm^ as the substrate and added ∆117Pro with or without VPg to monitor the possible VPg in the *cis* cleavage speed modulation of Pro. We observed that VPg did not increase the Pro cleavage speed. We also observed that Ca^2+^ and Zn^2+^ acted as inhibitors, whereas Mg^2+^ only slightly reduced Pro activity (Appendix A). Calcium is an essential nutrient involved in growth and development and is an important secondary messenger molecule. The Ca^2+^ under normal conditions is in the range of 10–200 nM; whereas the concentration of Ca^2+^ in the cell wall, vacuole, endoplasmic reticulum, and mitochondria is 1–10 mM, during stress, it can reach the micromolar level [43]. As viral infection can trigger plant stress signaling pathways, an increase in Ca^2+^ levels could be an indirect defense mechanism against viral infection. We hypothesize that this mechanism could have a favorable effect on viral particle maturation, as sobemovirus CP binds Ca^2+^ for particle stabilization [44]. In addition, Zn is involved in plant responses to pests and diseases [45]. As sobemovirus P1 is classified as a Zn-finger protein [46], it acts as a suppressor and activator of gene silencing [47] and is an infectivity factor involved in both viral replication and spreading [48], where the P1 Zn binding site may function as a redox sensor of the plant redox status to adapt its functions [49]. An increase in both elements in plant cells likely deactivates Pro, restricting polyprotein processing and reducing viral gRNA and sgRNA replication. However, the toxic effects of Ca^2+^ and Zn^2+^ can lead to cell death, thereby favoring systemic viral infection. Nevertheless, our data demonstrate that the presence or absence of RGMoV VPg does not have such a dramatic influence on *cis* proteolytic activity, as was the case with SeMV. We cannot exclude the modulating influence of RGMoV VPg on Pro activity *in planta*. However, complete loss of *cis* activity was not observed.

It has been noted previously that VPg presence acts as a switch between *in cis* and *in trans* active states [14,32]. Structural changes in sobemovirus Pro during interaction with VPg have been previously analyzed only with the CD spectrum by mutating key residues [15]. A more recent NMR study demonstrated that the interaction of VPg and Pro domains in SeMV is driven by a patch of hydrophobic residues on the surface of VPg and Pro domains [20]. Crystallography allowed us to identify visual changes in Pro structure during the interaction with VPg. We determined that a 10-residue-long VPg motif could assume the structure of an α-helix and bind to a specific site in the Pro domain. This interaction triggers major structural changes in the catalytic triad and substrate binding sites. These changes could certainly be the underlying mechanism for activation or at least stabilization of the *trans* activity of RGMoV Pro, but this mechanism is not universal among all sobemoviruses. Similar major structural changes have not been observed for the closely related SeMV Pro, the *in cis* active structure, which is very similar to VPg bound, a completely in *trans*-active RGMoV Pro. Although the interaction between Pro and VPg is driven by hydrophobic interactions in both RGMoV and SeMV, the responsive VPg structures—α-β-β-β domain for SeMV Pro and single α helix for RGMoV—are completely different. The RGMoV VPg motif WAD, with the key residue W^260^ according to PsiPred [50] and AlphaFold [51], is located in a helix, but in the SeMV case, the corresponding residues are located in a coil (Appendix A). The previously described essential W^43^ for SeMV VPg function [14,15] is located in different structural element strands, and there is no analog W residue for RGMoV in the corresponding region. Nevertheless, it can be concluded that both RGMoV and SeMV VPg structures contain an α-β-β-β domain in their *N*-terminal region, but their *C*-terminal parts after this domain are much more divergent, with RGMoV VPg containing several additional predicted α-helices and SeMV VPg being much more disordered. Overall, our research has identified novel interacting surfaces and key residues for VPg interaction with Pro, and demonstrated that the nature of this interaction and its implications are not conservative and could be highly different among sobemoviruses. The true meaning of the RGMoV Pro interaction with VPg remains controversial, since it is not directly involved in controlling the catalytic activity of Pro *in cis*, but is required *in trans*. Further studies are required to confirm this hypothesis.

Surprisingly, conformational differences were observed between the Δ117Pro, Δ117Pro-E/A-VPg, and Δ117Pro^cm^ states, with the only difference being the S^159^/A mutation. Different crystallization conditions could be at play, but it is very likely that these differences are actual representations of natural conformational oscillations in the Pro molecule.

## 4. Materials and Methods

### 4.1. Gene Amplification and Cloning

The cDNA clone of RGMoV MAFF. No. 307043 from Japan (GenBank: EF091714; [4]) was used as the source of the DNA matrix. As sobemovirus Pro at its *N*-terminal part contains a TMD sequence, for soluble RGMoV Pro expression in *E. coli* [12], we created an *N*-terminally truncated Pro, which lacks the first 50 aa (Δ50Pro) and contains a 6xH-tag at the *N*-terminus originating from the chosen expression vector [17]. The PCR product of Δ50Pro was first cloned into the cloning vector pTZ-57 using the InsTAclone PCR Cloning Kit (Thermo Fisher Scientific, Waltham, MA, USA). Clone selection with the corresponding insert was performed by restriction analysis and verified by Sanger sequencing with a BigDye Cycle Sequencing Kit and ABI Prism 3100xl Genetic Analyzer (Applied Biosystems, Waltham, MA, USA). This technique was used for all PCR products. Δ50Pro in the pETDuet1 (Novagen, Madison, WI, USA) expression vector was cloned through *Nco*I and *Hind*III cloning sites to create the pETDu-Δ50Pro plasmid. Clone selection was performed using a restriction analysis. The Δ50Pro-E/A-VPg fusion cleavage site mutant (Δ50Pro-E/A-VPg) was created as previously described for SeMV with site direct mutagenesis [14]: E^315^ and E^318^ (aa counting from RGMoV ORF2a first M) were mutated to A using oligonucleotides RG-SP-VPg-E/A-F and RG-SP-VPg-E/A-R, and RG-SerP-RAW-NdeI-F and RG2-VPg-HindIII-R (Appendix A) to introduce cloning sites. Both possible cleavage sites were mutated to prevent unexpected VPg cleavage from Pro. The cloning strategy was as in the Δ50Pro case only, as the expression vector was pET28a(+) (Novagen, Madison, WI, USA) using the *Nde*I and *Hind*III cloning sites, where the 6xH-tag at the *N*-terminus of Pro originated from a plasmid (pET-Δ50Pro-E/A-VPg).

The Δ50Pro-E/A-VPg and P16 fusion vectors with a 6xH-tag at the *C*-terminus (Δ50Pro-E/A-VPg-P16-C6H) were created by cloning the VPg-P16 domain with 6xH-tag at the *C*-terminus (VPg-P16-C6H) PCR product which was created by introducing the oligonucleotides RG2-VPg-NcoI-F and RG-P16-C6H-HindIII-R (Appendix A) into pET-Δ50Pro-E/A-VPg through the *Eco*RI and *Hind*III restriction sites (pET-Δ50Pro-E/A-VPg-P16-C6H) or into pET28a(+) using the *Nco*I and *Hind*III cloning sites for VPg-P16 fusion expression (pET-VPg-P16-C6H). The Δ50Pro-VPg native fusion with a 6xH-tag at the VPg *C*-terminus (Δ50Pro-VPg-C6H) was amplified using the oligonucleotides RG-SerP-RAW-NcoI-F and VPg-C-6-His-HindIII-R (Appendix A) and cloned into pET28a(+) using *Nco*I and *Hind*III cloning sites (Δ50Pro-VPg-C6H). The Δ50Pro TMD Pro cleavage site mutant was created by site-directed mutagenesis, where E^117^ was changed to A (Δ50Pro-TMD-E/A) using overlapping oligonucleotides RGSP-E/A-F and RGSP-E/A-R (Appendix A). An expression vector, pCold1-Δ50Pro (constructed earlier [17]), was introduced using the *Nde*I and *Xho*I restriction sites (pCold1-Δ50Pro-TMD-E/A). For the TMD Δ50Pro-E/A-VPg mutant (Δ50Pro-TMD-E/A-VPg), the pColdI-Δ50Pro-TMD-E/A vector was used, and *Cfr*42I and *Hind*III restriction sites were created after ligation of the pColdI-Δ50Pro-TMD-E/A-VPg plasmid. In the VPg-P16-C6H cleavage site mutant, E^393^ was mutated to A (VPg-E/A-P16-C6H) by overlapping oligonucleotides RG-VPg-E/A-P16-F and RG-VPg-E/A-P16-R (Appendix A) in the first PCR reaction and the second RG2-VPg-NcoI-F and RG-P16-C6H-HindIII-R (Appendix A), and the mutant was cloned into pET28a(+) using *Nco*I and *Hind*III cloning sites, which resulted in the pET-VPg-E/A-P16 expression vector. For Δ23Pro-C6H Pro-C-6-His-HindIII-R and RG-SerP-E/S-Nco-F, oligonucleotides (Appendix A) were used. *Δ23Pro-C6H* was cloned into the pTZ57 vector for sequence verification, and the correct *Δ23Pro-C6H* coding sequence was cloned into pET28a(+) using *Nco*I and *Hind*III cloning sites to obtain plasmid pET-Δ23Pro-C6H. The Δ50Pro-NC6H construct was created based on pCold1-Δ50Pro, where a *6xH-tag* sequence was added from pET-Δ23Pro-C6H in cloning sites *Xho*I and *Hind*III, creating the construct pCold1-Δ50Pro-NC6H.

The catalytic center Δ117Pro mutant (Δ117Pro^cm^) was created using oligonucleotides SP-S/A-F and SP-S/A-R (Appendix A), where S^159^ was substituted with A by site direct mutagenesis, and using SP-NCE-HindIII-R and SP-AAV-NcoI-N6H-F (Appendix A), cloning sites *Nco*I and *Hind*III were introduced for cloning into the pET28a(+) expression vector (pET-Δ117Pro^cm^-N6H).

### 4.2. Protein Expression and Purification

The C2566 (New England Biolabs, Ipswich, MA, USA) or BL21(DE3) (New England Biolabs, Ipswich, MA, USA) cell strains were used for all protein expression and co-expression assays. Overnight cultures grown at 37 °C were transferred to 200 mL 2xTY media at a ratio of 1:10. The cells were cultivated at 30 °C and 200 rpm until the optical density at 600 nm reached 0.8. Cells were induced with a final concentration of 0.2 mM IPTG and cultivated at 20 °C at 200 rpm for 16 h. After cultivation, the cells were collected by slow-speed centrifugation and frozen at −70 °C.

Cells were disrupted by ultrasound using an ultrasound disintegrator UP200S (Hielscher Ultrasonics, Teltow, Germany) at period 0.5 and intensity 70% for 16 min in disruption buffer (Δ50Pro, Δ50Pro-E/A-VPg, Δ117Pro^cm^, Δ50Pro-NC6H, Δ50Pro-VPg-C6H, Δ50Pro-E/A-VPg-P16-C6H, Δ50Pro-TMD-E/A, Δ50Pro-TMD-E/A-VPg, Δ23Pro-C6H, VPg-P16-C6H, VPg-E/A-P16-C6H, and all co-expressions 1xPBS; Δ50Pro^cm^ 0.5 M UREA, 1xPBS, 1 M NaCl). Δ50Pro^cm^ was purified on Protino Ni-IDA 2000 Packed Columns (Macherey-Nagel, Düren, Germany), and elution fractions containing Δ50Pro^cm^ were dialyzed in a 6–8 kDa dialysis membrane (Spectrum Chemical, New Brunswick, NJ, USA) against 1:100 volumes of 0.5 M UREA, 1xPBS, 1 M NaCl, and 5 mM imidazole. Purification of Δ50Pro, Δ50Pro-E/A-VPg, Δ117Pro^cm^, VPg-C6H, and P16-C6H was performed in two steps. First, we used PrepEase Histidine-tagged Protein Purification Midi Kit-High Yield (Affymetrix, Santa Clara, CA, USA) or Protino Ni-IDA 2000 packed columns (Macherey-Nagel, Düren, Germany) for Δ50Pro and Δ50Pro-E/A-VPg to facilitate the removal of the proteolytically cleaved *N*-terminal region of Pro. Gel filtration was then performed during the final cleaning step. The gel filtration process was performed on an Äkta Pure 25 XK 16/70 column packed with 120 mL Superdex 200 (GE Healthcare, Chicago, IL, USA), and purified proteins were eluted with 1xPBS at 1 mL/min by collecting 2 mL per fraction. Fractions containing purified protein were united and concentrated on Amicon-Ultra-15 10 K filtration units (Merck-Millipore, Darmstadt, Germany), and the Amicon-Ultra-0.5 10 K filtration units (Merck-Millipore, Darmstadt, Germany) were used for buffer exchange from 1xPBS to 20 mM TRIS-HCl (pH 8.0), suitable for protein crystallization. Protein concentrations were measured using a Qubit 2.0 Fluorometer with a Qubit Protein Assay Kit (Thermo Fisher Scientific, Waltham, MA, USA), according to the manufacturer’s protocol.

Cells from co-expression experiments were treated as previously described. Soluble proteins were purified with the PrepEase Histidine-tagged Protein Purification Midi Kit-High Yield (Affymetrix, Santa Clara, CA, USA) or Protino Ni-IDA 2000 packed columns (Macherey-Nagel, Düren, Germany). Protein concentrations were measured on a Qubit 2.0 (Thermo Fisher Scientific, Waltham, MA, USA) with a Qubit Protein Assay Kit (Thermo Fisher Scientific, Waltham, MA, USA), according to the provided protocol.

### 4.3. Protein Analysis on SDS-PAGE and Western Blotting (WB)

Samples were analyzed on 12.5% SDS-PAGE or NuPAGE™ 4–12%, Bis-Tris, 1.0 mm mini protein gels (Thermo Fisher Scientific, Waltham, MA, USA). The samples for gel analysis were diluted with a 1xLaemmly sample buffer (50 mM Tris pH 7.0, 1% SDS, 50% glycerin, 5% β-mercaptoethanol (β-ME), 0.004% bromophenol blue) and heated for 10 min at 95 °C. Samples on SDS-PAGE were visualized by Coomassie blue R-250 [10% ethanol, 10% glacier acid, 0.1% R-250 (Sigma-Aldrich, St. Louis, MO, USA)] or G-250 [20% ethanol, 2% trichloroacetic acid, 0.5 g/L Coomassie G-250 (Sigma-Aldrich, St. Louis, MO, USA)] staining.

WB was performed on a semi-dry Western blot device transferred to an Amersham Protran 0.45 mm nitrocellulose (NC) blotting membrane (GE Healthcare, Chicago, IL, USA) at 52 mA for 45 min. For a protein with 6xH-tag detection on the NC membrane, the H-Tag Antibody HRP Conjugate Kit (Merck-Millipore, Darmstadt, Germany) was used according to the manufacturer’s protocol.

### 4.4. Protein Analysis by Mass Spectrometry

Samples for mass spectrometry analysis were prepared as follows: 2 µL of purified protein (1 mg/mL) was mixed with 2 µL of 2% trifluoroacetic acid and 2 µL 2,5-dihydroxyacetophenone (2,5-DHAP; Bruker Daltonics, Leipzig, Germany) matrix solution (50 μmol 2,5-DHAP dissolved in 96% ethanol and 10 μmol aqueous diammonium hydrogen citrate), and 1 µL was spotted onto an MTP Anchor Chip 400/384TF (Bruker Daltonics, Leipzig, Germany) and left to crystallize. The sample analysis was performed on an “Autoflex” MALDI-TOF MS (Bruker Daltonics, Leipzig, Germany). A mix of mass calibration standards I and II or peptide molecular mass calibration standard II (Bruker Daltonics, Leipzig, Germany) was used for mass calibration.

### 4.5. In Vitro Δ50Pro^cm^, Δ117Pro and VPg Cleavage Test

For the in vitro cleavage experiments, all proteins were used in equal micromolar amounts (1:1:1). The micromolar concentrations were calculated according to Δ50Pro^cm^ (13.7 µM) in a 100 µL reaction volume. These results were confirmed by a time-course experiment in which samples were withdrawn from the reaction mixture at different time points (10 min, 1 h, 2 h, 3 h, and 10 h) at 37 °C, and immediately stopped with 4xLaemmli sample buffer (50 mM Tris pH 7.0, 4% SDS, 50% glycerin, 10% β-ME, 0.004% bromophenol blue) followed by sample heating at 95 °C for 10 min. Samples were analyzed in 15 well 1.0 mm Bolt 4–12% Bis-Tris Plus mini protein gels (Thermo Fisher Scientific, Waltham, MA, USA) and stained with Coomassie blue G-250.

### 4.6. Crystallization, Data Collection, and Structure Determination

Δ117Pro, Δ117Pro-E/A-VPg, and Δ117Pro^cm^ were transferred to 20 mM Tris-HCl (pH 8.0) and concentrated to 9.38 mg/mL, 2.9 mg/mL, and 18.1 mg/mL concentrations, respectively. All Pro variants were crystallized using the sitting drop method in 0.4 + 0.4 μL or 1 + 1 μL setups. The initial screening was performed using JCSG+, Structure screens 1 and 2, and Pact Premier screens (Molecular Dimensions, Rotherham, UK). Δ117Pro crystals grew within 2–3 weeks, and the final crystallization conditions were 0.1 M HEPES (pH 7.5), 22.5% *m*/*v* PEG3350, 0.22 M LiSO_4_ and 3% DMSO. Δ117Pro^cm^ crystals grew within one week, and the final crystallization conditions were 0.18 M ammonium sulphate, 22% PEG8000, and 0.1 M sodium cacodylate (pH 6.5). Δ117Pro-E/A-VPg crystals grew within 3–4 months in 0.1 M MIB, pH 9.0, and 25% PEG1500 conditions. Crystals were soaked in a cryoprotectant (30% glycerol in mother liquid) and frozen in liquid nitrogen. Diffraction data were collected at the MX 14.1 beamline in the BESSY II synchrotron electron storage ring operated by the Helmholtz-Zentrum Berlin (Berlin, Germany, [52]). Δ117Pro crystals were diffracted to a 2.3 Å resolution, Δ117Pro^cm^ crystals to a 1.6 Å resolution, and Δ117Pro-E/A-VPg to a 2.1 Å resolution.

The data were indexed with MOSFLM [53] and scaled using SCALA [54] from the CCP4 suite [55]. The Δ117Pro crystals had space group P2 with two chains in the asymmetric unit. The Δ117Pro-E/A-VPg crystals had a space group P2_1_2_1_2_1_ with one chain in the asymmetric unit. The Δ117Pro^cm^ crystals had a space group C2 with one chain in the asymmetric unit.

The Δ117Pro-E/A-VPg structure was determined by molecular replacement using MOLREP [56], using a Δ117Pro homology model built with SWISS-MODEL [57] from the Pro domain of SeMV (PDB ID 1ZYO) [32]. The VPg GEMTWADMVE helix was built by identifying the matching tryptophan residue in the VPg sequence, and built further by adding individual residues. The Δ117Pro and Δ117Pro^cm^ structures were determined in MOLREP using coordinates from the determined Δ117Pro-E/A-VPg structure. The models were further built manually with Coot [58], and refined with REFMAC5 [59] and phenix.refine [60]. Data processing, refinement, and validation statistics are shown in Appendix A. All structures were deposited in the worldwide Protein Data Bank under accession numbers 6FEZ (Δ117Pro), 6FF0 (Δ117Pro-E/A-VPg), and 7YZV (Δ117Pro^cm^).

## Figures and Tables

**Figure 1 ijms-24-05347-f001:**
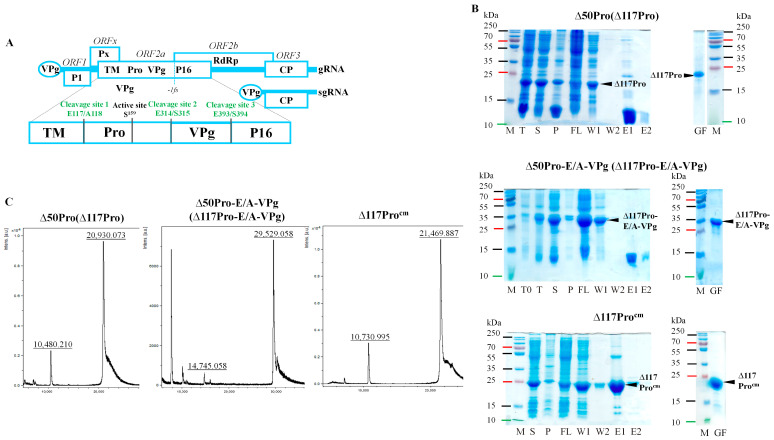
Δ50Pro, Δ50Pro-E/A-VPg, and Δ117Pro^cm^ expression and analysis. (**A**)—RGMoV schematic overview; number at aa—location in ORF2a starting from the first M; TM: transmembrane domain; Pro: protease; VPg: virus-genome liked protein; P16: protein with 16 kDa MW; RdRp: RNA dependent RNA polymerase; (**B**)—SDS-PAGE analysis of expression and purification of Δ50Pro, Δ50Pro-E/A-VPg, and Δ117Pro^cm^; M—protein marker (Page Ruler Plus, Thermo Fisher Scientific, Waltham, MA, USA); T0— total cell lysate before expression; T—total cell lysate after expression; S—soluble protein fraction; P—pellet; FL—flow fraction after IMAC column; W1—first wash; W2—second wash; E1—first elution fraction; E2—second elution fraction; GF—sample after gel-filtration and concentration; (**C**)—mass spectrometer analysis.

**Figure 2 ijms-24-05347-f002:**
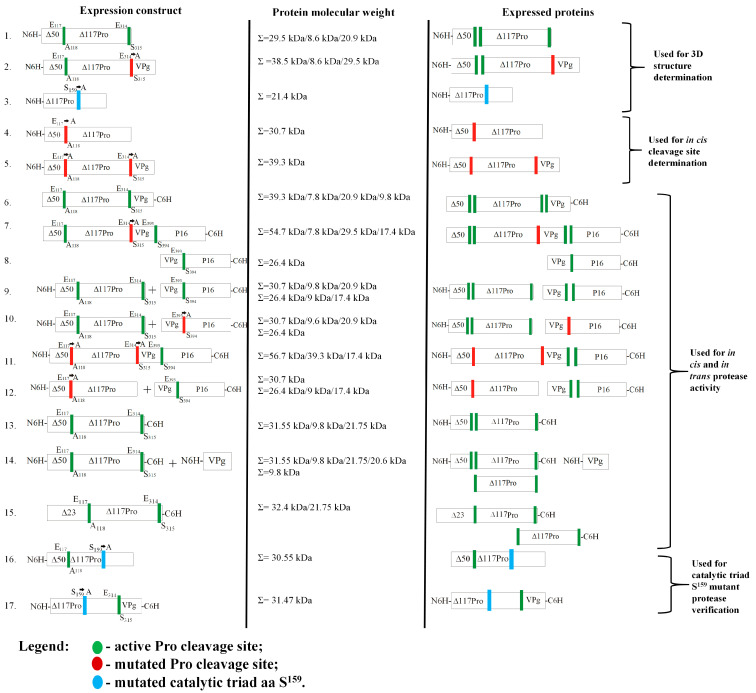
Schematic overview of the Pro variants used for 3D structure determination, catalytic activity experiments depending on active cleavage sites and cofactors or catalytic triad amino acid (aa) residues. 1—**Δ50Pro**, active cleavage site E^117^/A^118^ *in cis*, active cleavage site E^314^/S^315^ *in trans*, 6xH-tag at *N*-terminus; 2—**Δ50Pro-E/A-VPg**, active cleavage site E^117^/A^118^ *in cis*, inactivated cleavage site E^314^/S^315^ by mutation of E to A, 6xH-tag at the *N*-terminus of Pro; 3—**Δ117Pro^cm^**, catalytic triad S^159^ mutated to A; 4—**Δ50Pro-TMD-N6H**, inactivated cleavage site E^117^/A^118^ by mutation of E to A, active cleavage site E^314^/S^315^ *in trans*, 6xH-tag at *N*-terminus of Pro; 5—**Δ50Pro-TMD-E/A-VPg-N6H**, inactivated cleavage site E^117^/A^118^ by mutation of E to A, inactivated cleavage site E^314^/S^315^ by mutation of E to A, 6xH-tag at *N*-terminus of Pro; 6—**Δ50Pro-VPg-C6H**, active cleavage site E^117^/A^118^ *in cis*, active cleavage site E^314^/S^315^ *in trans*, 6xH-tag at *C*-terminus of VPg; 7—**Δ50Pro-E/A-VPg-P16-C6H**, active cleavage site E^117^/A^118^ *in cis*, inactivated cleavage site E^314^/A^315^ E mutation of E to A, active cleavage site E^393^/S^394^ *in trans*, 6xH-tag at *C*-terminus of P16; 8—**VPg-P16-C6H**, active cleavage site E^393^/S^394^ *in trans*, 6xH-tag at *C*-terminus of P16; 9—**Δ50Pro-N6H** co-expression with **VPg-P16-C6H**; 10—**Δ50Pro-N6H** co-expression with **VPg-E/A-P16-C6H**, inactivated cleavage site E^393^/S^394^ by mutation of E to A, 6xH-tag at *C*-terminus of P16; 11—**Δ50Pro-TMD-E/A-VPg-P16-CN6H**, inactivated cleavage site E^117^/A^118^ by mutation of E to A, 6xH-tag at *N*-terminus of Pro, inactivated cleavage site E^314^/S^315^ by mutation of E to A, active cleavage site E^393^/S^394^ *in trans*, 6xH-tag at *C*-terminus of P16; 12—**Δ50Pro-TMD-N6H** co-expression with **VPg-P16-C6H**; 13—**Δ50Pro-NC6H**, active cleavage site E^117^/A^118^ *in cis*, active cleavage site E^314^/S^315^ *in trans*, 6xH-tag at *N*- and *C*-terminus; 14—**Δ50Pro-NC6H**, active cleavage site E^117^/A^118^ *in cis*, active cleavage site E^314^/S^315^ *in trans*, 6xH-tag at *N*- and *C*-terminus of Pro co-expression with **VPg-N6H**, 6xH-tag at *N*-terminus of VPg; 15—**Δ23Pro-C6H** active cleavage site E^117^/A^118^ *in cis*, active cleavage site E^314^/S^315^ *in trans*, 6xH-tag at *N*-terminus; 16—**Δ50Pro^cm^-N6H**—catalytic triad S^159^ mutated to A; 17—**Δ117Pro^cm^-VPg-NC6H**—catalytic triad S^159^ mutated to A, 6xH-tag at *N*-terminus of Pro, 6xH-tag at *C*-terminus of VPg. TMD: transmembrane domain; Δ50: *N*-terminally truncated by 50 aa; Pro: protease; VPg: virus-genome liked protein; P16: protein with 16 kDa molecular weight; red bare: represents inactivated Pro sites; green bare: represents active Pro cleavage sites; blue bar: represents catalytic triad S; box: represents protein domains.

**Figure 3 ijms-24-05347-f003:**
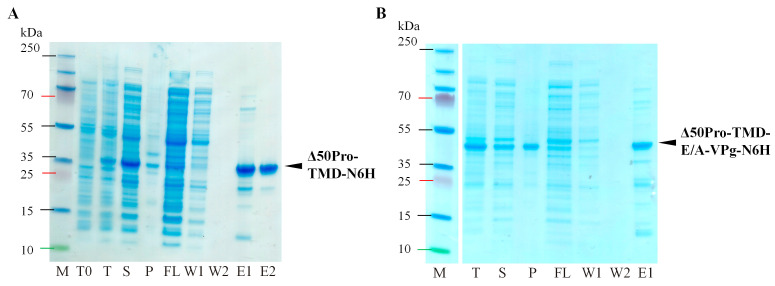
Δ50Pro-TMD-N6H and Δ50Pro-TMD-E/A-VPg-N6H expression purification analysis. (**A**)—expression and purification analysis of Δ50Pro-TMD-N6H; (**B**)—expression analysis of Δ50Pro-TMD-E/A-VPg-N6H; M—protein marker (Page Ruler Plus, Thermo Fisher Scientific, Waltham, MA, USA); T0—total cell lysate before expression; T—total cell lysate after expression; S—soluble protein fraction; P—pellet; FL—flow fraction after IMAC; W1—first wash; W2—second wash; E1—first elution fraction; E2—second elution fraction; arrow indicates expressed Pro.

**Figure 4 ijms-24-05347-f004:**
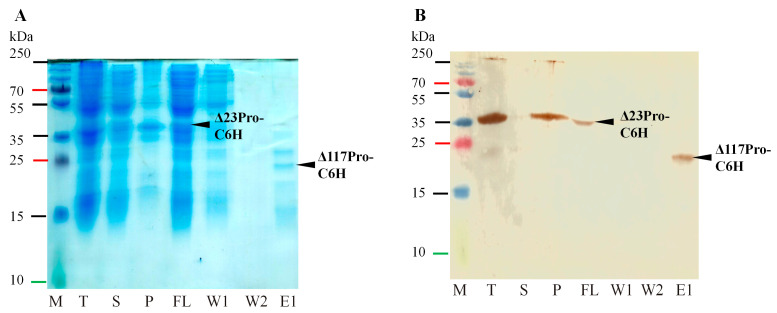
Δ23Pro-C6H expression and purification analysis in SDS-PAGE and WB. (**A**)—expression and purification analysis of Δ23Pro-C6H in SDS-PAGE; (**B**)—expression and purification analysis of Δ23Pro-C6H in WB with 6xH-tag antibodies in ratio 1:1000; M—protein marker (Page Ruler Plus, Thermo Fisher Scientific, Waltham, MA, USA); T—total cell lysate after expression; S—soluble protein fraction; P—pellet; FL—flow fraction after IMAC; W1—first wash; W2—second wash; E1—first elution fraction.

**Figure 5 ijms-24-05347-f005:**
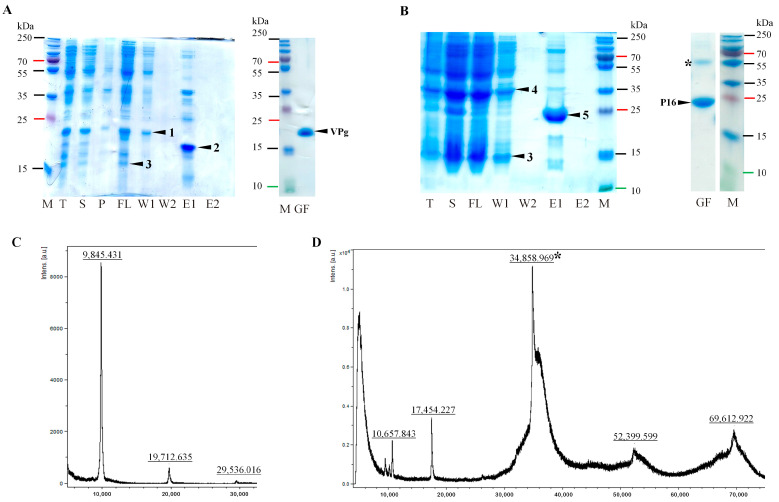
Expression and purification analysis of VPg-C6H and P16-C6H from polyprotein processing. (**A**)—expression and purification analysis of Δ50Pro-VPg-C6H in SDS-PAGE; (**B**)—expression and purification analysis of Δ50Pro-E/A-VPg-P16-C6H in SDS-PAGE; (**C**)—mass spectrometer analysis of VPg-C6H; (**D**)—mass spectrometer analysis of P16-C6H; M—protein marker (Page Ruler Plus, Thermo Fisher Scientific, Waltham, MA, USA); T—total cell lysate after expression; S—soluble protein fraction; P—pellet; FL—flow fraction after IMAC; W1—first wash; W2—second wash; E1—first elution fraction; 1—*C*-terminus of proteolytically truncated Δ50Pro (Δ117Pro); 2—VPg-C6H; 3—*N*-terminus of proteolytically truncated Δ50Pro; 4—*C*-terminus of Δ50Pro-E/A-VPg (Δ117Pro-E/A-VPg); 5—P16-C6H; GF—sample after gel filtration and concentration; *—P16-C6H dimer.

**Figure 6 ijms-24-05347-f006:**
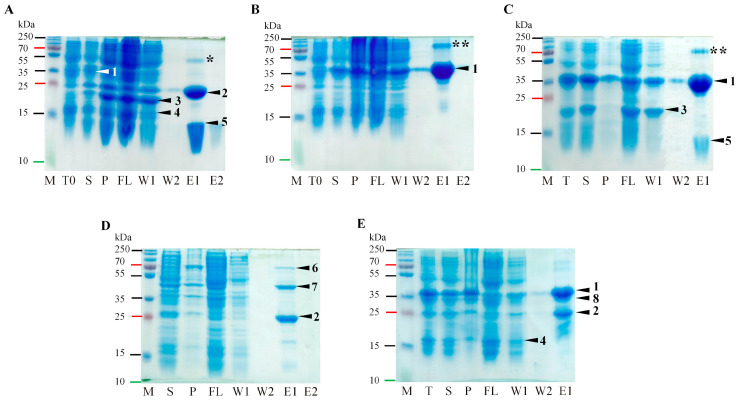
Pro *in trans* cleavage test and VPg-P16 cleavage site mapping. (**A**)—co-expression and purification analysis of Δ50Pro-N6H and VPg-P16-C6H in SDS-PAGE; (**B**)—expression and purification analysis of VPg-P16-C6H in SDS-PAGE; (**C**)—co-expression and purification analysis of Δ50Pro-N6H and VPg-E/A-P16-C6H in SDS-PAGE; (**D**)—expression and purification analysis of Δ50Pro-TMD-E/A-VPg-P16-NC6H; (**E**)—co-expression and purification analysis of Δ50Pro-TMD-E/A-N6H and VPg-P16-C6H in SDS-PAGE; M—protein marker (Page Ruler Plus, Thermo Fisher Scientific, Waltham, MA, USA); T—total cell lysate after expression; S—soluble protein fraction; P—pellet; FL—flow fraction after IMAC; W1—first wash; W2—second wash; E1—first elution fraction; 1—VPg-P16-C6H; 2—P16-C6H; 3—*C*-terminus of proteolytically truncated Δ50Pro; 4—VPg; 5—*N*-terminus of proteolytically truncated Δ50Pro; 6—Δ50Pro-TMD-E/A-VPg-P16-NC6H; 7—Δ50Pro-TMD-E/A-VPg; 8—Δ50Pro-TMD-N6H; *—P16-C6H dimer, **—VPg-P16-C6H dimer.

**Figure 7 ijms-24-05347-f007:**
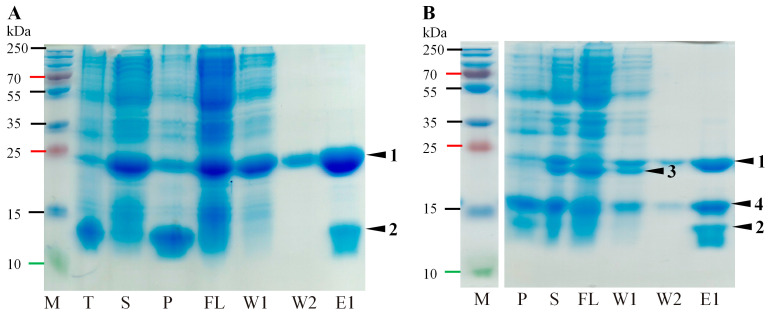
Pro *in cis* and *in trans* cleavage test without and with VPg-N6H. (**A**)—Δ50Pro-NC6H expression and purification analysis in SDS-PAGE; (**B**)—co-expression and purification analysis of VPg-N6H and Δ50Pro-NC6H in SDS-PAGE; M—protein marker (Page Ruler Plus, Thermo Fisher Scientific, Waltham, MA, USA); T—total cell lysate after expression; S—soluble protein fraction; P—pellet; FL—flow fraction after IMAC; W1—first wash; W2—second wash; E1—first elution fraction; 1—Δ117Pro-N6H; 2—*N*-terminus of proteolytically truncated Δ50Pro; 3—Δ117Pro; 4—VPg-N6H.

**Figure 8 ijms-24-05347-f008:**
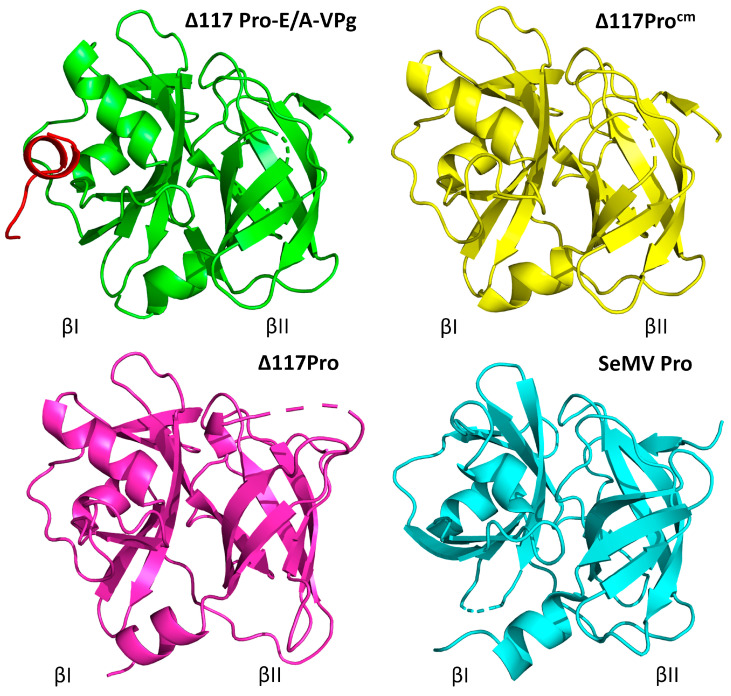
3D crystal structures of Δ117 Pro-E/A-VPg (PDB ID 6FF0, green) and VPg sequence GEMTWADMVE (red), Δ117Pro^cm^ (PDB ID 7YZV, yellow), RGMoV Δ117Pro (PDB ID 6FEZ, magenta), and SeMV Pro domain (PDB ID 1ZYO, cyan). Molecule sides containing βI and βII barrels are marked.

**Figure 9 ijms-24-05347-f009:**
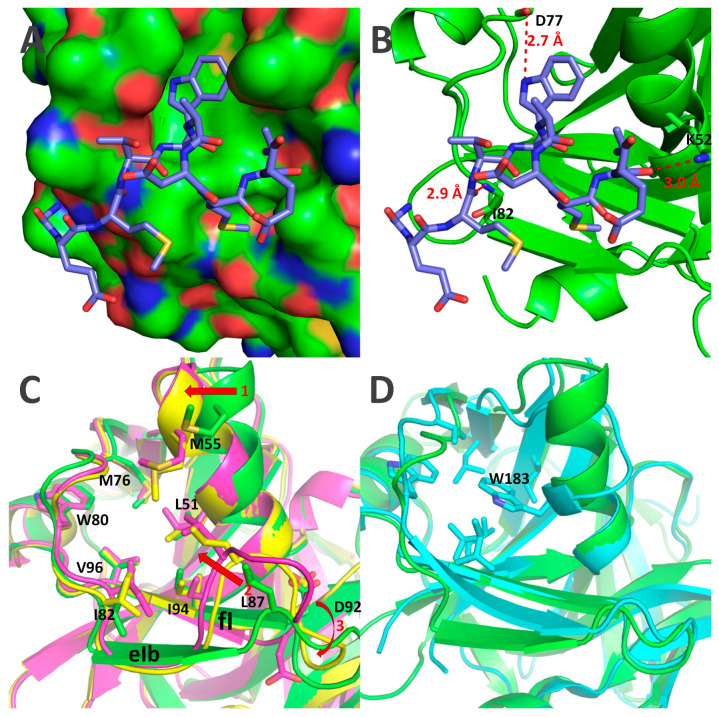
RGMoV VPg and its Δ117Pro binding site on βI barrel. (**A**)—a surface model of Δ117Pro-E/A-VPg with bounded VPg GEMTWADMVE motif, corresponding to 256–265 aa (stick model). In the surface model, the carbon atoms are displayed in green, oxygen atoms in red, and nitrogen atoms in blue. (**B**)—hydrogen bonds formed between VPg and Δ117Pro. Distances are measured in angstroms. (**C**)—aa forming RGMoV Pro VPg binding pocket and the conformational shifts in the structure of α1 helix (shift 1), in the eIb-fI loop (shift 2), and the positioning of catalytic triad D^92^ (shift 3). Shifts are indicated with red arrows. (**D**)—comparison of RGMoV Δ117Pro-E/A-VPg (PDB ID 6FF0, green) and SeMV Pro domain (PDB ID 1ZYO, cyan). The hydrophobic pocket of the SeMV Pro domain is indicated by stick modelling.

**Figure 10 ijms-24-05347-f010:**
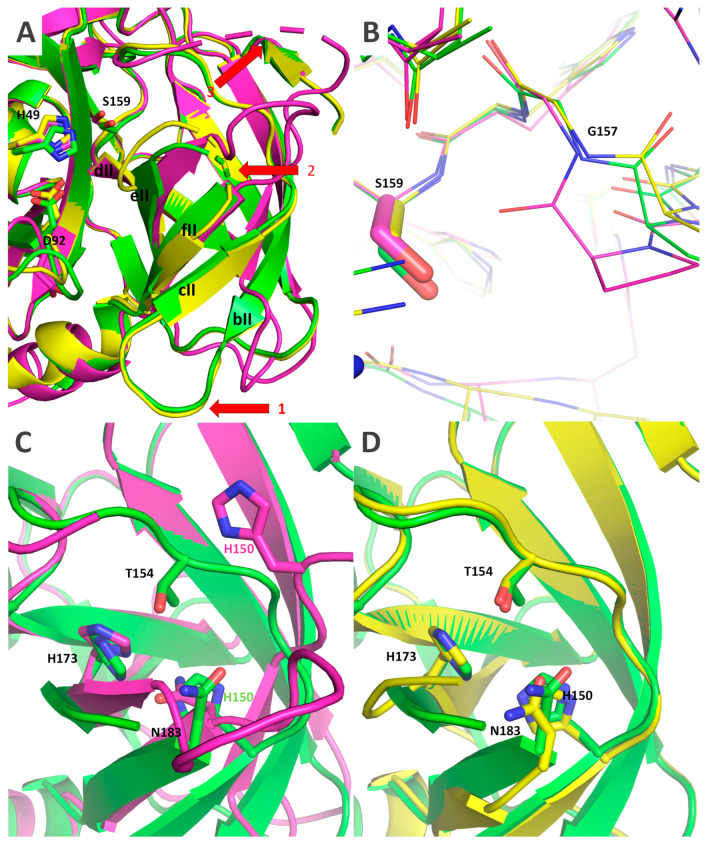
The conformational transition between RGMoV Δ117Pro VPg-bound and VPg-free states in βII barrel. (**A**)—conformational shifts of bII and cII β-strands (shift 1), eII/fII loop (2), and cII/dII loop (3) in Δ117Pro-E/A-VPg (PDB ID 6FF0; green), Δ117Pro (PDB ID 6FEZ, magenta), and Δ117Pro^cm^ (PDB ID 7YZV, yellow) structures. (**B**)—possible disruption of oxyanion hole formed by G^157^ in Δ117Pro (PDB ID 6FEZ, magenta) compared to Δ117Pro-E/A-VPg (PDB ID 6FF0, green) and Δ117Pro^cm^ (PDB ID 7YZV, yellow) structures. (**C**)—influence of the conformational shift in eII/fII and cII/dII loops on the positions conservative key residues in Glu-binding site (H^150^, H^173^, T^154^ and N^183^) in Δ117Pro (PDB ID 6FEZ, magenta) and Δ117Pro-E/A-VPg (PDB ID 6FF0, green) structures. (**D**)—influence of the conformational shift in eII/fII and cII/dII loops on the positions conservative key residues in Glu-binding site (H^150^, H^173^, T^154^ and N^183^) in Δ117Pro^cm^ (PDB ID 7YZV, yellow) and Δ117Pro-E/A-VPg (PDB ID 6FF0, green) structures.

**Figure 11 ijms-24-05347-f011:**
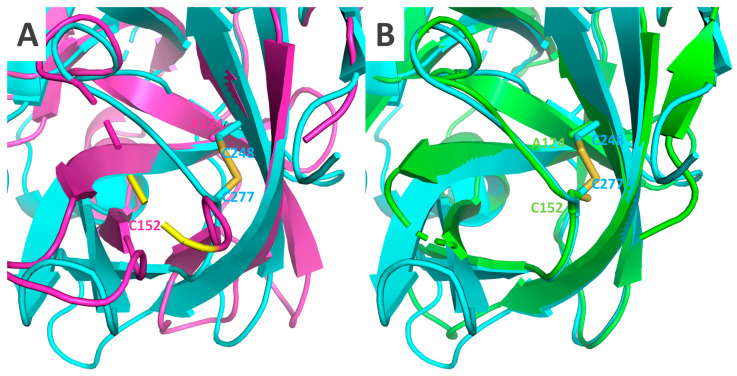
SeMV Pro domain (PDB ID 1ZYO, cyan) substrate binding site and C^277^-C^248^ disulfide bridge. SeMV Pro domain is aligned with Δ117 (magenta, (**A**)) and Δ117-Pro-E/A-VPg (green, (**B**)).

## Data Availability

The structures of RGMoV Pro variants were deposited in worldwide Protein Data Bank under accession numbers 6FEZ (Δ117Pro), 6FF0 (Δ117Pro-E/A-VPg), and 7YZV (Δ117Pro^cm^). The authors declare that the data supporting the findings of this study are available within the article and Appendix A.

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
