# Peer review of "VPg Impact on Ryegrass Mottle Virus Serine-like 3C Protease Proteolysis and Structure"

_ijms, 2023, doi:10.3390/ijms24065347_

Round 1
Reviewer 1 Report
In present study, the authors solved a full Pro-VPg 3D structure of ryegrass mottle virus (RGMoV) that demonstrates the structural changes in three different conformations due to VPg interaction with Pro. The authors also confirmed the existence of an unusual previously unmapped cleavage site (E/A) for sobemovirus Pro in the transmembrane domain. Overall, the manuscript is well organized and presented. However, several points should be addressed before the MS accepted for publication in International Journal of Molecular Sciences:
1) Line 19 and Line 80, the authors should add the full name of NMR.
2) Line 155, “B:” should be changed as “C:”.
3) Figure 1B, Its difficult to understand and distinguish each column in the figure of SDS-PAGE analysis.
4) Line 552, 560, 571, 582, 586, 589, and 593, Rstriction enzyme writing should be italicized.
5) The quality of figures in text is poor, need further revision.
6) Many editorial errors like above were also found in the MS and references. The authors should rewrite this MS carefully.
Author Response
Comments and Suggestions for Authors
In present study, the authors solved a full Pro-VPg 3D structure of ryegrass mottle virus (RGMoV) that demonstrates the structural changes in three different conformations due to VPg interaction with Pro. The authors also confirmed the existence of an unusual previously unmapped cleavage site (E/A) for sobemovirus Pro in the transmembrane domain. Overall, the manuscript is well organized and presented. However, several points should be addressed before the MS accepted for publication in International Journal of Molecular Sciences:
We greatly appreciate the Reviewer's work, and we are thankful for the appreciation of our work and suggestions on improving the manuscript.
1) Line 19 and Line 80, the authors should add the full name of NMR.
Thank You for the remark. We added the full name of the NMR in both indicated Lines.
2) Line 155, “B:” should be changed as “C:”.
Thank You for the remark. The “B” was changed to “C”.
3) Figure 1B, Its difficult to understand and distinguish each column in the figure of SDS-PAGE analysis.
We slightly modified Figure 1. Transferred Figure 1B to the right side of the figure and enlarged the gel images. Hopefully, now each column is distinguishable in the SDS-PAGE.
4) Line 552, 560, 571, 582, 586, 589, and 593, Rstriction enzyme writing should be italicized.
Our selected English language editing service – Editage, removed italicization from restriction enzymes writing, but we changed to italic again.
5) The quality of figures in text is poor, need further revision.
We slightly modified the figures and uploaded them in higher quality. Hopefully, the quality of the figures will stay the same after all journal editorial steps.
6) Many editorial errors like above were also found in the MS and references. The authors should rewrite this MS carefully.
Thank You for the remark. Errors occurred in references during the reference style conversation by EndNote. We apologise for this. We revisited references and the MS for error editing.

Reviewer 2 Report
In this MS the authors present their results about the protease activity of Pro of ryegrass mottle virus (RGMoV), a member of the Sobemovirus genus, in the presence and absence of the VPg. The presented results are really interesting as they decipher new protease properties of the RGMoV Pro compared to that of SeMV, as well as the role of VPg in the in cis and in trans activity of the protease. Overall the MS is well written, the results are well presented and supported from the provided data, therefore this MS merits publication in IJMS. Some minor comments are found in the attached file.

Author Response
Comments and Suggestions for Authors
In this MS the authors present their results about the protease activity of Pro of ryegrass mottle virus (RGMoV), a member of the Sobemovirus genus, in the presence and absence of the VPg. The presented results are really interesting as they decipher new protease properties of the RGMoV Pro compared to that of SeMV, as well as the role of VPg in the in cis and in trans activity of the protease. Overall the MS is well written, the results are well presented and supported from the provided data, therefore this MS merits publication in IJMS. Some minor comments are found in the attached file.
We thank the Reviewer for Your appreciation of our work, valuable suggestions on improving the manuscript quality and time spent on the review.
We revisited our MS according to respective Reviewer comments and updated Figure 1 according to the remark.
